# Deformation Monitoring and Shape Reconstruction of Flexible Planer Structures Based on FBG

**DOI:** 10.3390/mi13081237

**Published:** 2022-07-31

**Authors:** Huifeng Wu, Rui Dong, Zheng Liu, Hui Wang, Lei Liang

**Affiliations:** 1National Engineering Research Center of Fiber Optic Sensing Technology and Networks, Wuhan University of Technology, Wuhan 430070, China; huizifeng_829@163.com (H.W.); wanghui@whut.edu.cn (H.W.); 2School of Electronic Information and Automation, Guilin University of Aerospace Technology, Guilin 541004, China; Liuz@guat.edu.com; 3Engineering Comprehensive Training Center, Guilin University of Aerospace Technology, Guilin 541004, China; dongrui202112@163.com

**Keywords:** data-driven model, BP neural network, FBG grating array, shape reconfiguration

## Abstract

To reduce the dependence of real-time deformation monitoring and shape reconstruction of flexible planar structures on experience, mathematical models, specific structural curvature (shape) sensors, etc., we propose a reconstruction approach based on FBG and a data-driven model; with the aid of ANSYS finite element software, a simulation model was built, and training samples were collected. After the machine learning training, the mapping relationship was established, which is between the strain and the deformation variables (in three directions of the *x*-, *y*-, *z*-axis) of each point of the surface of the flexible planar structure. Four data-driven models were constructed (linear regression, regression tree, integrated tree, and BP neural network) and comprehensively evaluated; the predictive value of the BP neural network was closer to the true value (R^2^ = 0.9091/0.9979/0.9964). Finally, the replication experiment on the flexible planar structure specimen showed that the maximum predictive error in the *x*-, *y*-, and *z*-axis coordinates were 2.93%, 35.59%, and 16.21%, respectively. The predictive results are highly consistent with the expected results of flexible planar structure deformation monitoring and shape reconstruction in the existing test environment. The method provides a new high-precision method for the real-time monitoring and shape reconstruction of flexible planar structures.

## 1. Introduction

Deformation monitoring and shape reconstruction of large flexible planar structures are one focal point of current research, such as the wind power engine impeller, solar panel, aircraft wing, helicopter blade, and other structural monitoring, which is favored by many scholars. The research method is also becoming increasingly diverse; for example, the inverse finite element method (IFEM) [1,2,3], ko displacement theory [4,5], camera measurement method [6,7], fiber grating sensing measurement method [8,9,10,11,12,13] (FBG is a popular direction in current sensor research, and its applications are wide [14,15]), etc. Each of the above methods has its own characteristics [16]. The ko displacement theory is based on segmental linearization, which requires a large number of sensors to ensure the accuracy of the measurement. The camera measurement method requires high image clarity, resulting in a large amount of data by system processing and poor real-time monitoring, which is not suitable for real-time monitoring of large flexible planar structures. Relatively, the IFEM and fiber grating sensing measurement method are applied and promoted to a higher degree at present. Dong et al. [17] proposed an IFEM based on beam elements to reconstruct the shape of the main load-bearing structure of an adaptive deformable wing. Kefal et al. [18] applied the IFEM to the real-time displacement reconstruction of a wing sandwich structure with medium thickness through a strain sensor network and obtained 3D real-time deformations and strains in agreement with the DIC/FEM results. Wang et al. [19] proposed a pipeline deformation monitoring method using the IFEM based on iBeam3 units. The experimental results show that the method can effectively monitor the deformation of buried pipelines in the process of soil freezing and thawing with a measurement error of 15.43%. Gherlone et al. [2] proposed a method based on the IFEM using measured surface strain data to reconstruct the elastic dynamic structural response of trusses, beams, and frame structures. Oboe et al. [20] proposed an improved IFEM based on isogeometric analysis (IGA), which conducted centralized loads and distributed load tests on the wing structure and obtained more accurate and effective displacement reconstruction. Zhao et al. [21] proposed an improved IFEM based on IGA to perform concentrated and distributed load tests on the wing structure. The experimental results showed that the improved IFEM was more accurate and effective in displacement reconstruction. However, in the above documents, the choice of mathematical models and boundary conditions of the IFEM did not give a specific derivation process or the selection of standard descriptions, all of which were from personal experience. 

FBG is increasingly being used in shape reconstruction. Floris et al. [22] listed various specific structures of FBG shape sensors to bring significant advances in aerospace engineering and biomedical applications. Wu et al. [23] proposed an improved object shape reconstruction method based on a FBG shape sensor by embedding FBG arrays into silica gel to reconstruct the object shape. In 2012, Luna [24] used multicore optical fiber as a shape sensor to reconstruct the spatial shape of the airfoil surface with a reconstruction error of 1.5% or less. Wu et al. [25] designed a high-precision soft substrate shape sensor based on a dual fiber grating, which was able to improve the shape reconstruction accuracy with a reconstruction error of 6.13%. He et al. [26,27] designed two different types of curvature sensors using silicone and polyimide with FBG to complete the 3D shape reconstruction of the airfoil. Wang et al. [28] proposed a curvature sensor based on a polyimide sensing layer to achieve bending shape monitoring of soft surgical manipulators, and the experimental results showed that the maximum error between the measured value and the actual bending curvature was less than 2.1%. Fiber grating sensing measurement methods can achieve real-time measurements of flexible planar structures based on diverse fiber grating sensing forms [29,30,31,32], but there is a strong dependence on the specific structure curvature (shape) sensor, while the curvature (shape) sensor preparation error and calibration error also have an impact on the measurement accuracy of the system. 

The rapid development of artificial intelligence techniques has promoted the combination of structural deformation monitoring with machine learning, such as neural networks [33,34,35], decision trees, support vector machines, linear regression, and random forests. The above algorithms are used in conjunction with FBG to predict the displacement or load of flexible planar structures to improve the measurement accuracy with the help of big data-driven models. Kim et al. [36] used multiple linear regression and variance analysis to optimize the design of rotor shapes. Kaveh et al. [37] used different machine learning techniques to establish the relationship between the fiber angle and the flexural capacity of a cylinder under bending-induced loads to predict the final buckling load of the variable stiffness composite cylinder. Sefati et al. [38,39] investigated three data-driven models for tip position estimation (DPE) of surgical manipulators and compared DPE and shape reconstruction results, with the following maximum absolute errors observed for DPE: 0.78 mm and 2.45 mm for free bending motion and 1.22 mm and 3.19 mm obstacles for task space, indicating that, compared to conventional methods, the proposed data-driven method has superior performance. Lum et al. [40] studied robots using FBG to provide position information and machine learning to build models for real-time, robust, and reliable shape reconstruction. Alexakis et al. [41] provided a new approach to apply statistical modeling and machine learning to install a FBG network on a Victorian railroad viaduct to track train speeds and dynamic strain amplitudes of the bridge. Klotz et al. [42] developed a generalized deflection and twist measurement method for wings based on FBG and neural networks to achieve real-time wing deformation measurements on the ground and in the air. Li et al. [43] developed a model-free method based on neural networks to reconstruct the shape of a continuous robotic arm. The experimental results showed that the reconstruction accuracy was improved compared with the traditional method. 

There is much reference on the application of FBG combined with machine learning, but not much reference on the combination of deformation monitoring of large flexible planar structures with machine learning. In response to the problems of low accuracy, a large number of sensors, and serious dependence on specific structural curvature (shape) sensors or personal experience, this paper proposes a method of flexible structural deformation measurement based on FBG and machine learning to reconstruct flexible structural shapes in real time. Different from most reconstruction methods of flexible planar structures based on FBG curvature (shape) sensors, the proposed method is based on deformation variables of measuring points on the flexible structure surface to complete the shape reconstruction of the structure. To achieve the method, four FBG grating arrays are set on the surface of the flexible structure to obtain the strain values of measuring points on the structure surface from the FBG grating. Using machine learning, the model of the strain value and the deformation variables of measuring points are trained and established. The deformation variables (in the *x*-axis, *y*-axis, *z*-axis direction) of each measuring point on the structure surface are predicted to obtain the real-time three-dimensional coordinate value of each measuring point. Finally, by an improved Kalman filter algorithm, the 3D shape of the flexible structure deformed by bending and twisting is reconstructed. 

We theoretically change the traditional working model of flexible planar structure deformation monitoring, verify the linear relationship between the strain and deformation of the measuring points and collect a large number of training data samples by using ANSYS finite element software, introduce the data-driven concept, and establish the mapping relationship between the strain and deformation of measuring points through machine learning methods, which increases the dependence of the monitoring system on the training data, reduces the dependence of flexible structural deformation monitoring on specific structural curvature (shape) sensors, and reduces the influence of empirical factors such as mathematical formulas and boundary conditions on shape reconstruction. By replicating the contents of the simulation tests on small flexible plate specimens, we verified the feasibility of the proposed method and the accuracy of the predicted values of the machine learning model with predictive error values that match the existing test environment. 

Therefore, the main contributions of this paper are as follows. 

(1)Breaking through traditional mode-driven thinking, the development of a design framework for monitoring the deformation of flexible planar structures based on the combination of FBG arrays and the machine learning algorithm. The 4 × 4 FBG grating arrays are used to monitor the strain values of measuring points of flexible planar structures in real time. The machine learning algorithm predicts the deformation variables of each measuring point.(2)A system simulation model is established by ANSYS finite element software to provide a large amount of training data for machine learning, and the mapping relationship between the strain and deformation variables of each measuring point on a flexible flat structure is established. (3)Machine learning algorithms such as the multiple linear regression model, regression tree model, integrated tree (bagging tree) model, and BP neural network are listed and compared for the prediction of the deformation variables of measuring points; the BP neural network obtains higher accuracy for structural shape reconstruction due to the coefficient of determination R^2^ of 0.9091/0.9979/0.9964. (4)The introduction of the data-driven model increases the dependence of the monitoring system on the obtained training dataset and replicates experiments to demonstrate the feasibility and reliability of the proposed method, which achieves more stable and reliable shape reconstruction without either customizing the FBG curvature (shape) sensors for various structures or relying on an empirical selection of functional relationships compared to the references [26,27].

## 2. FBG Sensor Network and System Simulation

### 2.1. FBG Sensing Principle

FBG is also known as fiber Bragg grating; its essence is an optical fiber with a periodic change in the core refractive index, that is, the grating with a periodic distribution of spatial phases formed within the fiber core; its role is to form a narrow band within the core (transmission or reflection) filter or mirror. As shown in Figure 1 [44].

For standard single-mode fiber grating, the central wavelength of the reflected light depends on the effective refractive index and the grating period, which satisfies the following relationship:(1)$$\lambda_{B}=2n_{eff}\Lambda$$
where: $\lambda_{B}$ is the reflected light center wavelength; $n_{eff}$ is the effective refractive index of the fiber core; Λ is the grating period. 

$n_{eff}$ and Λ are functions of the temperature and strain. This dependence [45] is described as shown in Equation (2).
(2)$$\frac{\Delta \lambda_{B}}{\lambda_{B}}=\left(1-P_{e}\right)\varepsilon +\left(\alpha +\xi \right)\Delta T$$
where: $P_{e}$—the photoelastic parameter of the optical fiber core for a general single-mode fiber $P_{e}\approx 0.22$.

α—the thermal expansion of the optical fiber core.

ξ—the thermal-optic coefficient of the optical fiber core.

FBG is considered to measure strain; the default temperature change is 0, so Equation (2) simplifies to Equation (3).
(3)$$\varepsilon =\frac{\Delta \lambda_{B}}{\lambda_{B}}\times \frac{1}{\left(1-P_{e}\right)}$$

*ε* is the measured point strain value when the flexible planar structure is subjected to the bending moment and torque.

### 2.2. FBG Sensing Network

To verify the proposed method of deformation monitoring and shape reconstruction, we chose a 300 mm (L) × 200 mm (W) × 2 mm (H) 304 stainless steel plate as the specimen of the flexible planar structure and used the workbench module of ANSYS finite element software to simulate the strain and deformation variable of each measuring point on the structural specimen under different loads to form a training dataset with a complete characteristic quantity and identification. 

The shape reconstruction method we discuss is mainly used for the shape reconstruction of flexible structures such as wind power engine impellers, solar sails, aircraft wings, helicopter rotor systems, and other propellers; the simplified structure with one end fixed and the other end free is used for simulation verification as shown in Figure 2. There are a total of 16 (4 × 4) measuring points on the structural specimen to form a small FBG sensing network, which consists of four parallel optical fibers with a 50 mm fiber spacing, each of which is engraved with four different wavelength gratings, and the grating interval is 50 mm. The long side of the structural specimen is the *x*-axis; the width is the *y*-axis; the blue dot is the coordinate origin; and the coordinates of the nearest grating measuring point from the coordinate origin are (20, 0, 0) (Remark: Due to the simple structure of the test piece, the sensors are placed in a uniform distribution; for other large flexible structures or complex flexible structures, the placement of sensors needs to be considered by the optimization algorithm).

### 2.3. Measurement System Simulation

The simulation model of the system shown in Figure 3a is established by the workbench module of ANSYS finite element software. The material properties of the structural specimen are shown in Table 1. For flexible planar structures, there are two main types of loads that have a significant impact on the structural deformation: torque and the force in the vertical direction. Differences in the loading value or loading point will lead to differences in structural deformation. 

When the torque is 10 N·m, the coordinates of its loading point are (1,199,0); the force is 31,867 N, and the coordinates of its loading point are (1,100,0). The total deformation cloud of the test part structure is shown in Figure 3c.

The system adopts the parametric modeling method to complete the simulation of the deformation and stress of structural specimens under different loads. The parameters include two types, namely, input parameters and output parameters. The input parameters are the load information. If the direction of force is vertical up, the value is positive; otherwise, the value is negative. If the direction of torque is counterclockwise, the value is positive; otherwise, the value is negative, as shown in Table 2. The output parameters include the normal elastic strain of 16 measuring points and the deformation variables in the *x*, *y* and *z* directions (The deformations considered in this paper are all within the range of elastic deformations, and inelastic deformations are discussed separately).

The combinations of input parameters with different values are randomly generated by MATLAB, which are input to the ANSYS simulation model to simulate and generate the corresponding output parameters of each measuring point.

## 3. Machine Learning Approaches

### 3.1. Preparing Dataset

Dataset preparation is one of the key tasks in machine learning, which involves data collection, data organization, data preprocessing, feature engineering and labeling, and finally division of the data into the training set and test set.

**Data collection**: Three thousand input data samples are input into the ANSYS simulation model, which are divided into three categories: 1000 data samples are loaded with vertical force alone; 1000 data samples are loaded with torque alone; and 1000 data samples are loaded with force and torque simultaneously. The 3000 simulation model output data samples are collected as the training dataset for machine learning.

**Data organization**: The objective of this paper is to obtain the deformation variable of each measuring point in the case of a known strain value of the measuring point. This problem is formulated as a regression problem, where the strain value of each measuring point is the independent variable, and the deformation variable of the measuring point is the dependent variable.

**Data splitting**: In the process of training the model, the splitting rate of 75–25% is used for data splitting [44], which has been widely used in machine learning model training and has shown good training performance, i.e., 75% of the dataset is used as training data, and 25% of the dataset is used as test validation data.

### 3.2. Machine Learning (ML) Model Selection

Commonly used supervised learning regression models include linear regression, regression tree, support vector regression (SVR), k-nearest neighbor, random forest, and neural network models, of which the latter four models can perform both classification and regression. We choose four of them, namely, the multiple linear regression model, regression tree model, integrated tree (bagging tree) model, and BP neural network model, which tend to produce more accurate predictions compared to other machine learning regression models [44].

In each of the 3000 samples, the independent variable is the strain value of 16 measuring points, and the dependent variable is the output deformation variable (*x*, *y*, *z* directions) of 16 measuring points. To test the predictive performance of the model, the deformation variables of measuring point 1 in the *x*, *y*, and *z* directions are selected for model training. The hold-out method is used to verify the model, namely, 25% of the sample data are used to test the model-fitting effect. The effects of the true value and the predictive value and the corresponding residual are shown in Figure 4 and Figure 5.

The output responses in Figure 4 show that the distribution area of the prediction values of the linear regression model appears to overlap with the distribution area of the true values, and the two distribution regions of the regression tree model and the integrated tree model have a significant difference. In the first 100, 400th–550th samples, the difference between the two distribution areas of the neural network model obviously appears between the 400 and 550 samples. The response diagram can show the correlation between the prediction values and the true values of the models but does not reflect the prediction performance of each model.

The residual is the difference between the predictive value and true value of the model. From the distribution of the residuals in Figure 5, it can be seen that the linear regression residuals are more active in the whole sample interval, especially in the first 100 samples; the linear regression model, regression tree model, and integrated tree model residuals are active, and the residuals exhibited by the BP neural network are obviously more convergent than the other three models.

In machine learning, linear regression model evaluation usually uses the mean squared error (MSE), root mean square error (MSER), and coefficient of determination (R^2^) as performance parameters to evaluate the model predictive effect quantitatively. The evaluation parameters of the predictive model of measuring point 1 in the three axes are shown in Table 3, Table 4 and Table 5.

From Table 3, Table 4 and Table 5, the RMSE and MSE values of the four types of regression models approach 0, which means that the prediction effect of each model is equivalent, but from the R^2^ consideration, the prediction performance of each model is as follows: the regression tree < integrated tree < linear regression < BP neural network, and the prediction value of the BP neural network model is closer to the true value. In the comprehensive consideration, the proposed method uses a BP neural network as the data-driven training model.

### 3.3. BP Neural Network Training Regression Model

Using the powerful function of the BP neural network, the strain variables of 16 measuring points are selected as input, and the deformation variables of the *z*-axis direction of 16 measuring points are used as the prediction output to perform model training (the prediction method of the *x*- and *y*-axis direction variables is the same).

The network parameters are set as follows.

The learning factor is 1.0 × 10^−^^2^; the number of training iterations is 1000; the minimum error of the training target (error function target value) is 1 × 10^−5^; the activation function is the sigmoid function shown in Equation (4)
(4)$$\mathrm{sigmoid}(x)=\frac{1}{1+e^{-x}}$$

Using the Levenberg–Marquardt optimization algorithm in the gradient descent method, the training results are shown in Figure 6. Figure 6 shows the response between the predicted and true values of the deformation variables in the *z*-direction of the 16 measuring points, and Figure 6 shows the residuals between the predicted and true values of deformation variables in the *z*-direction of the 16 measuring points.

The response figure reflects the feasibility of the neural network training model, and the distribution areas of the predicted and true values of the 16 measuring points overlap. The residual figure reflects the accuracy of the model prediction. The maximum residual value appears in the combination of residuals of the 9th measuring point, taking the value of 5.45248 × 10^−3^, and the second is 6.46454 × 10^−3^ at the 5th measuring point. The maximum true value of the 9th measuring point is 6.63287 × 10^−2^, and the maximum true value of the 5th measuring point is 9.17637 × 10^−2^. Then, the maximum percentages of residuals are 8.22% (9th measuring point) and 7.04% (5th measuring point), which are within the acceptable error range. The data fitting results are shown in Figure 7.

The coefficient of determination of the training model is also called the goodness of fit. The larger the coefficient of determination is, the higher the degree of explanation of the dependent variable to the independent variable, the higher the percentage table of changes caused by the independent variables to the total changes, and the closer the coefficient of determination is to 1, the better the fit performance of the model and the closer the predictive value is to the true value.

During neural network model training, the MSE is used to quantify the trend of the error function during the continuous iterations, and it can be found from Figure 8 that the MSE gradually decreases during the iteration process, except for individual outliers that cannot be eliminated, which shows that the model converges during the training process.

## 4. Experiments and Results

To verify the proposed method of flexible planar structure deformation monitoring and shape reconstruction, the experimental platform shown in the simulation system was replicated as shown in Figure 9.

As shown in Figure 9, the test consists of a computer, an eight-channel FBG high-speed wavelength demodulator, FBG grating sensing network, and flexible structural specimen. The structural specimen is made of 304 stainless-steel with the same size as the ANSYS finite element simulation software, in which one end is fixed, and three holes are drilled in the other free end. The three holes are the “Loading points”, which used to hang mass blocks of different weights to simulate different loading states. The “Measuring points” are randomly selected FBG grating sensing network to compare the predicted values with the actual values, The FBG grating sensing network is composed of four FBG grating arrays. Each grating array connects four FBG sensors in series, and one FBG sensor represents a measuring point. The grating array is produced by Zhongshan Jingliang Optoelectronics Technology Co., Ltd(Zhongshan City, China). Taking the third grating array in Figure 9 as an example, the FBG sensor parameters are shown in Table 6 below.

The experimental content is carried out in two conditions. One is gradually to load the mass blocks, which are 250 g, 500 g, 1500 g, at the loading point in the middle of the free end of the structural specimen, which makes the structural specimen bend and deform, as shown in Figure 10a. The improved Kalman filter algorithm is used to reconstruct the 3D shape of the structural specimen in MATLAB software, as shown in Figure 11a. The other is to load different loads at the loading points on both sides of the free point of the structural specimen. One loading point is suspended with a mass block, and the other loading point is pulled up, which continuously loads three states, as shown in Figure 10b. The three-dimensional shape reconstruction of the structural specimen is shown in Figure 11b.

Any two points on the structural specimen are selected to compare the coordinate values obtained by different measuring methods in different states to evaluate the model prediction accuracy. The measuring points are shown as green circles in Figure 9.

Due to the limited experimental environment and equipment, the true value is measured by micrometers, and the *y*-axis coordinate values do not change when the flexible planar structure is bent and deformed during loading of the middle loading point. The value of the *x*-axis, *y*-axis, and *z*-axis coordinate changes when the flexible planar structure is deformed by the torque. Table 7 and Table 8 show that the larger the deformation variables of the measuring point are, the higher the accuracy of the predictive result. The maximum predictive error of the *x*-axis coordinate value is 2.93%; the maximum predictive error of the *y*-axis coordinate value is 35.59%; and the maximum predictive error of the *z*-axis coordinate value is 16.21%, so the predictive accuracy is relatively high, especially the prediction of the *x*-axis coordinate value. The predicted results are highly consistent with the expected results under the existing test environment and equipment.

## 5. Discussion

We propose a method for measuring the deformation of flexible planar structures based on FBG and machine learning and verify the feasibility and reliability of the method from both theoretical and experimental aspects in this paper. 

**In the model simulation**, Figure 2, Figure 3 and Figure 4 show that there are linear relationships between each point strain value and variables on the flexible planar structure. Additionally, Table 3, Table 4 and Table 5 show that, in the case of known strain variables of the measuring points, the comprehensive coefficient of determination R^2^ = 92.29% reflects that the predictive accuracy of the BP neural network model outperforms that of the multivariate linear fit model, regression tree model, and integrated tree model.

**In the replication experiment,** Table 7 and Table 8 show that the maximum error of the predicted coordinate value of the measuring points, *x*-axis < *z*-axis < *y*-axis, especially the deformation variables in the *z*-direction of the measuring points, are the maximum, and the maximum error of the predicted coordinate value in the *z*-axis direction is 16.21%. The flexible planar structure will be slightly deformed in the *y*-axis direction due to the torque, and the predicted error of the coordinate value will be increased because the deformation variable is too small. The minimum predicted error in the *y*-axis direction is 8.22%, but, overall, all the errors are within an acceptable range. In the references [27,46], the authors’ team pasted 48 FBGs onto a polyimide film to form a flexible wing smart skin, calculated the bending curvature of the polyimide skin, and reconstructed the 3D shape of the polyimide skin with different wing profiles. Compared with the 3D vision, the maximum error in the *z*-axis was less than 5%. However, only one state of bending deformation is considered in the reference, and the 3D reconstruction of the action of the torque was not described. The maximum predicted error in the *z*-axis under bending deformation is 4.93% in this paper. In the reference [17], the authors used the IFEM based on beam elements to reconstruct the main load-bearing structure of a deformable wing with a fishbone shape, and the deformation of the fishbone structure was mainly displacement along the *z*-axis. The total error of all strain measuring points on the fishbone structure did not exceed ±3 mm; the error at the tip was 3 mm; and the global maximum displacement reconstruction error was less than 8.62%. This reference was based on the free tetrahedral mesh of COMSOL for finite element simulation analysis of the fishbone structure of the adaptive deformable wing. The simplification of the model, the derivation of the formula, and the selection of the boundary conditions were not given a specific derivation process but came more from experience.

In a comprehensive comparison, the reconfiguration method proposed in this paper is based on data-driven thinking modes to establish a training framework of the flexible structural measuring point strain—deformation variables based on FBG and machine learning—to obtain measuring point deformation variables from measuring point strain characteristic information, which can dilute the dependence of the monitoring system on empirical and mathematical models and reduce the dependence of the system on specific structural curvature sensors. The advantages of the reconstruction method proposed in this paper are as follows.

(1). The proposed method can not only solve the shape reconstruction of flexible planar structures due to bending deformation but also the shape reconstruction of the deformation generated by torque. At present, most studies regarding the reference of flexible structural deformation only consider the shape reconstruction of bending deformation, such as in reference [46,47], so the method proposed in this paper improves the accuracy of shape reconstruction from the type of deformation.

(2). The introduction of the data-driven model breaks through the traditional shape reconstruction thought process. The traditional flexible planar structure shape reconstruction is mostly used with mode-driven thought processes, whether it is based on curvature shape reconstruction or inverse finite element reconstruction methods, all of which are secondary developments based on the experience or functional relationships accumulated in the early stage. For example, the reference [25] used shape reconstruction based on FBG curvature sensors by first determining the curvature in FBG curvature sensors as a function of the wavelength drift of the center of the FBG and then considered the reconstruction relationship between curvature and shape later.

(3). Finally, the BP neural network training model is selected to establish the prediction model of the strain and deformation of the measuring points, and the prediction accuracy of the model has a large impact on the accuracy of shape reconstruction. When the BP neural network uses the sigmoid function as the activation function, any function can approach a three-layer network with arbitrary accuracy. The BP neural network model trained in this paper maintains low deviation and variance, as shown in Figure 6, and has good predictive performance.

However, in the replication experiment, the predictive errors in the three axes caused by the torque are quite different. In particular, the maximum predictive error on the *y*-axis is 35.59%. There are two main reasons for this.

(1). Due to limited experimental conditions, the error caused by the tools and methods of measurement has a greater impact, especially the deformation in the *y*-axis, which is small, and the error brought by the measurement is larger. In the follow-up, a 3D high-definition camera or total station should be used as the actual value measurement tool.

(2). In the training samples, the *y*-axis deformation is small, and the true value of the output is small, such as loading a 360 N vertical downward force at the loading point on the left side of the flexible planar structure. The deformation variable of measuring point 1 in the *y*-axis is 6.78 × 10^−7^ m. The predictive accuracy will be amplified in the process of performing the conversion of the measuring point coordinate values. In the follow-up training model, for the case where there are small values of data in the training samples, it is considered to use a comprehensive training model to improve the prediction accuracy.

## 6. Conclusions

In this paper, we propose a training model framework for flexible planar structures measuring point strain—the deformation variable, based on FBG and machine learning. Using this framework, we predict the deformation variables of each measuring point on the flexible planar structure in the *x*-, *y*-, and *z*-axes and then use the improved Kalman filter algorithm to complete the shape reconstruction. Comparing and analyzing the multiple linear regression model, regression tree model, bagging tree model, and neural network for predicting the performance, the neural network is more suitable for such large flexible planar structure deformation monitoring and the prediction of deformation variables. The final replication experiment on the flexible planar structure specimen showed that the maximum predictive error value of the *x*-axis coordinate is 2.93%; the maximum predictive error value of the *y*-axis coordinate is 35.59%; and the maximum predictive error value of the *z*-axis coordinate is 16.21%. The predictive results are highly consistent with the expected results of the flexible planar structure deformation monitoring and shape reconstruction under the existing test environment. The method provides a new high-precision method for the real-time monitoring and shape reconstruction of flexible planar structures, such as wind power engine impellers, solar panels, aircraft wings, and helicopter blades.

## Figures and Tables

**Figure 1 micromachines-13-01237-f001:**
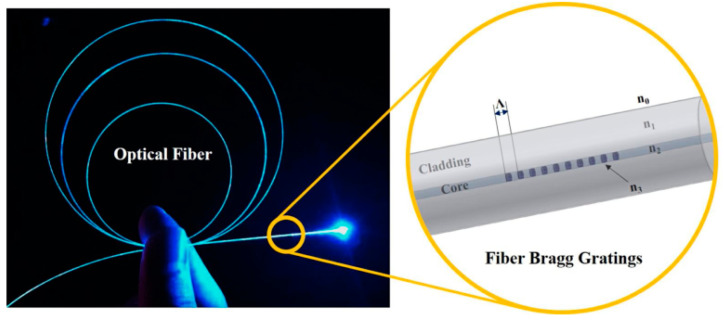
Details of fiber grating coil.

**Figure 2 micromachines-13-01237-f002:**
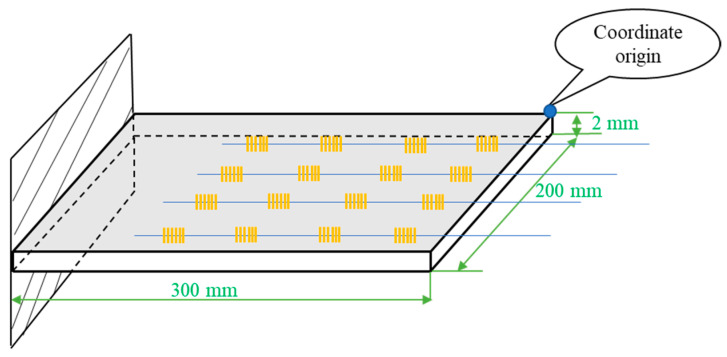
Sensor layout diagram.

**Figure 3 micromachines-13-01237-f003:**
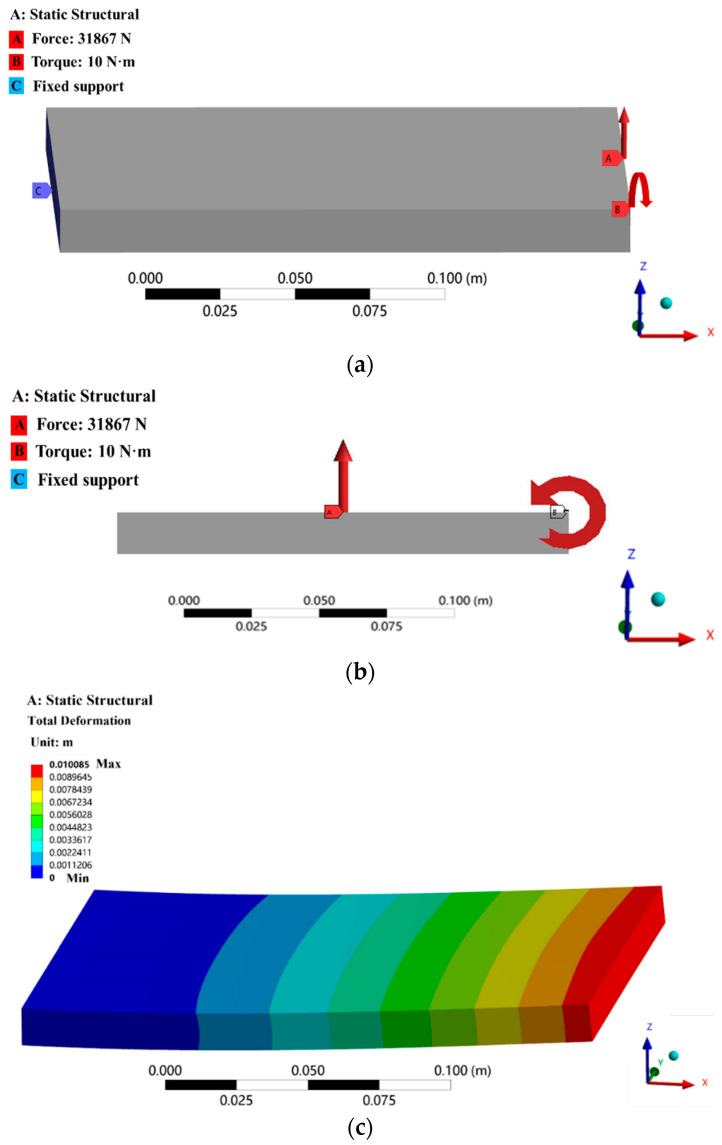
System simulation process: (**a**) Simulation model loaded state; (**b**) Simulation model loaded state (Left view); (**c**) Total deformation cloud of simulation model.

**Figure 4 micromachines-13-01237-f004:**
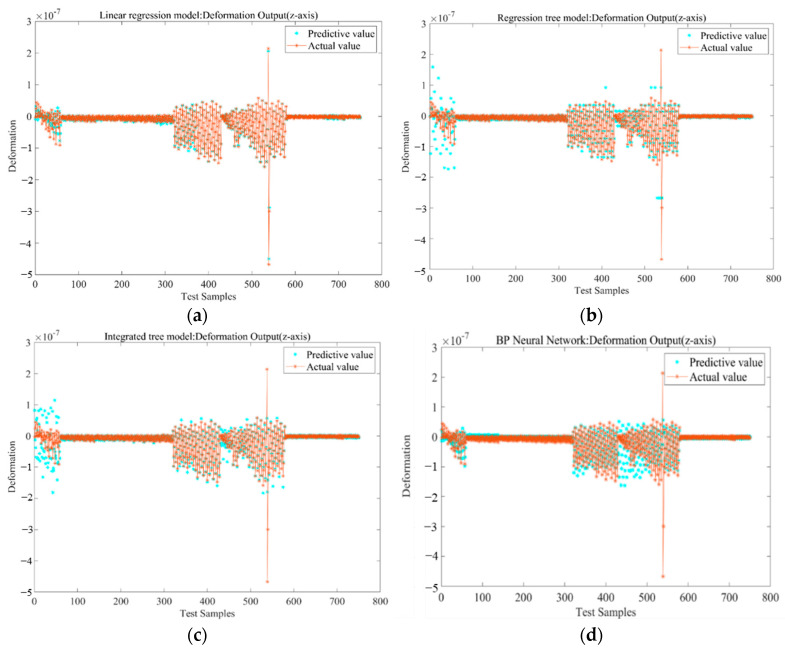
Output response of each training model: (**a**) Linear regression model; (**b**) Regression tree model; (**c**) Integration Tree Model; (**d**) BP neural network model.

**Figure 5 micromachines-13-01237-f005:**
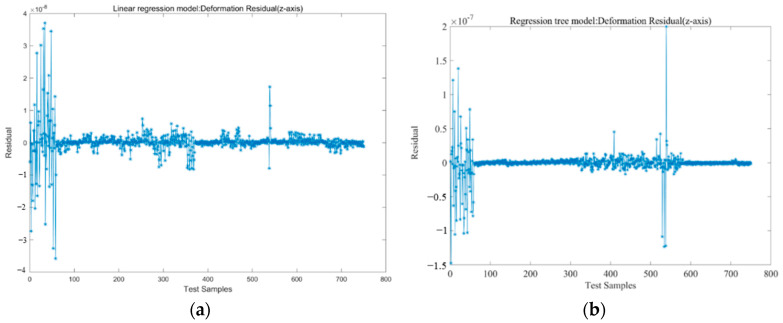
Residual of each training model: (**a**) Linear regression model; (**b**) Regression tree model; (**c**) Integration Tree Model; (**d**) BP neural network model.

**Figure 6 micromachines-13-01237-f006:**
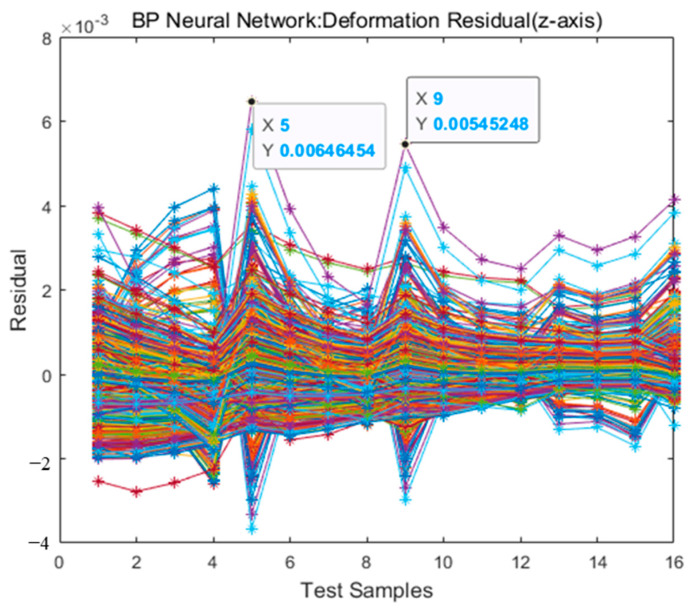
Residual between predictive value and true value.

**Figure 7 micromachines-13-01237-f007:**
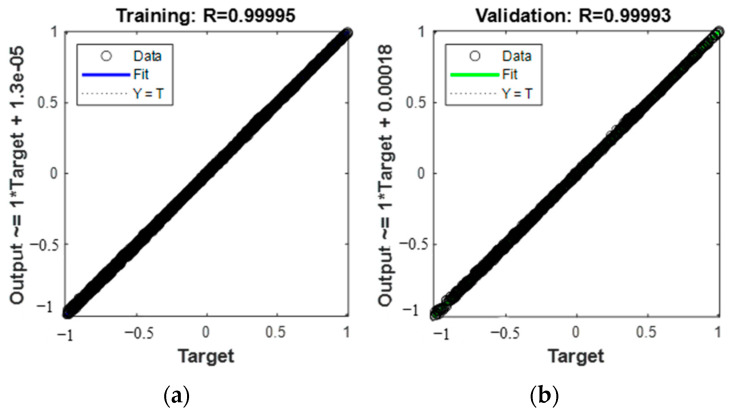
Neural network prediction correlation effect graph; (**a**) training sample correlation; (**b**) validation sample correlation; (**c**) test sample correlation; (**d**) combined correlation.

**Figure 8 micromachines-13-01237-f008:**
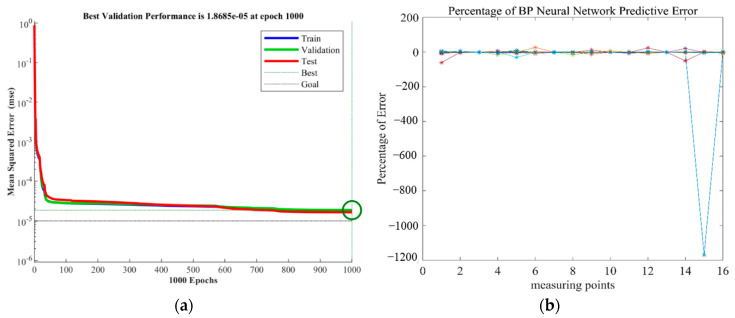
Model training process: (**a**) Relationships between the MSE of neural network and epochs; (**b**) Percentage of prediction error of measurement points.

**Figure 9 micromachines-13-01237-f009:**
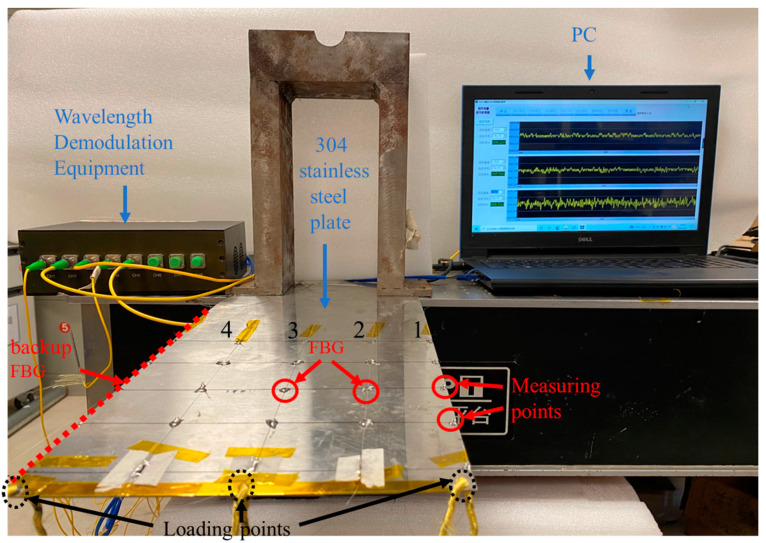
Testing System for Deformation Monitoring and Shape Reconfiguration of Flexible Structural.

**Figure 10 micromachines-13-01237-f010:**
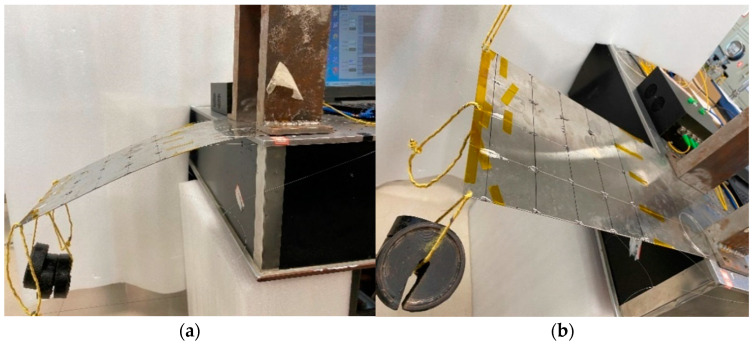
Load loading method of the structural specimen: (**a**) Hanging mass block at middle loading point; (**b**) different loads on both sides loading points.

**Figure 11 micromachines-13-01237-f011:**
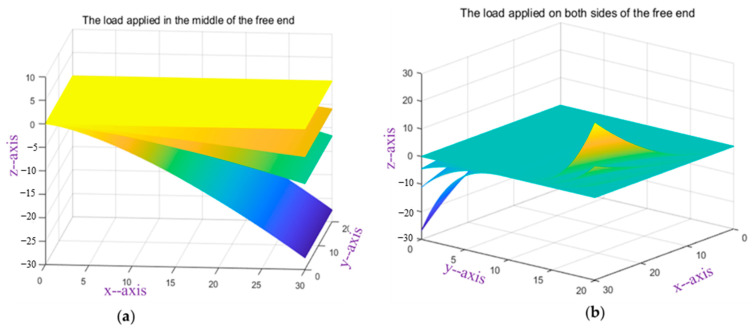
Shape Reconfiguration Chart of the structural specimen: (**a**) Hanging mass block at middle loading point; (**b**) different loads on both sides loading points.

**Table 1 micromachines-13-01237-t001:** The material properties of the structural specimen.

Property Category	Value
Density	7.93 g/cm^3^
Yield Strength	205 MPa
Elastic modulus	502 MPa
Tensile strength	199 GPa
Poisson’s ratio	0.3

**Table 2 micromachines-13-01237-t002:** Input parameter information table.

Parameters	Value	Unit
Force *y*-axis	Between 0 and 200	mm
Force *x*-axis	Between 0 and 300	mm
Force	Random	N
Torque *y*-axis	Between 0 and 190	mm
Torque *x*-axis	Between 0 and 290	mm
Torque	Random	N·m

**Table 3 micromachines-13-01237-t003:** Measuring point 1 (*z*-axis) regression model evaluation parameters.

Model Types	RMSE	R^2^	MSE
the linear regression	4.5380 × 10^−9^	0.4008	6.2139 × 10^−8^
regression tree model	1.9042 × 10^−8^	0.4777	2.6075 × 10^−7^
the integrated tree	2.5230 × 10^−8^	0.8128	3.4547 × 10^−7^
BP neural network	3.5377 × 10^−8^	0.9091	4.8442 × 10^−7^

**Table 4 micromachines-13-01237-t004:** Measuring point 1 (*x*-axis) regression model evaluation parameters.

Model Types	RMSE	R^2^	MSE
the linear regression	2.1077 × 10^−6^	0.9581	2.8861 × 10^−5^
regression tree model	2.8776 × 10^−6^	0.9581	2.8776 × 10^−5^
the integrated tree	2.0963 × 10^−6^	0.9581	2.8705 × 10^−5^
BP neural network	6.8763 × 10^−6^	0.9979	9.4157 × 10^−7^

**Table 5 micromachines-13-01237-t005:** Measuring point 1 (*z*-axis) regression model evaluation parameters.

Model Types	RMSE	R^2^	MSE
the linear regression	3.5601 × 10^−9^	0.9919	4.8749 × 10^−8^
regression tree model	2.694 × 10^−8^	0.6718	3.6889 × 10^−7^
the integrated tree	1.9725 × 10^−8^	0.5741	2.7010 × 10^−7^
BP neural network	1.2567 × 10^−8^	0.9964	1.7208 × 10^−7^

**Table 6 micromachines-13-01237-t006:** Parameters of the FBG Sensors.

	FBG1	FBG2	FBG3	FBG4
Grating type	Single-mode	Single-mode	Single-mode	Single-mode
Grating length (mm)	5	5	5	5
Central wavelength (nm)	1529.953	1533.146	1538.119	1544.385
Reflectivity (%)	75	75	75	75

**Table 7 micromachines-13-01237-t007:** Figure 11a comparison of measuring point results.

	Results	Measuring Point 1		Measuring Point 2	
		*x*-Axis	*z*-Axis	*x*-Axis	*z*-Axis
State I	predictive value(cm)	27.473	−5.329	22.118	−4.109
	true value(cm)	27.241	−5.128	22.367	−3. 916
	error %	0.85	3.92	1.11	4.93
State II	predictive value(cm)	26.897	−10.037	21.869	−8.046
	true value(cm)	26.675	−10.234	21.688	−8.257
	error %	0.83	1.92	0.83	2.56
State III	predictive value(cm)	24. 245	−20.396	20.792	−16.449
	true value(cm)	24. 441	−20.773	20. 965	−16.844
	error %	0.8	1.63	0.83	2.35

**Table 8 micromachines-13-01237-t008:** Figure 11b comparison of measuring point results.

	Results	Measuring Point 1			Measuring Point 2		
		*x*-Axis	*y*-Axis	*z*-Axis	*x*-Axis	*y*-Axis	*z*-Axis
State Ⅳ	predictive value(cm)	27.495	0.11	−3.668	22.688	0.038	−2.524
	true value(cm)	27.142	0.088	−3.879	22.043	0.059	−2.172
	error %	1.3	25	5.44	2.93	35.59	16.21
State Ⅴ	predictive value(cm)	26.144	0.135	−8.289	21.817	0.091	−5.825
	true value(cm)	26.597	0.121	−8.078	22.134	0.122	−6.103
	error %	1.7	11.57	2.61	1.43	25.41	4.56
State Ⅵ	predictive value(cm)	24.141	0.201	−17.003	20.104	0.15	−11.852
	true value(cm)	24.716	0.222	−17.413	20.418	0.187	−11.378
	error %	2.33	9.46	2.35	1.54	19.79	4.17

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
