# Peer review of "Deformation Monitoring and Shape Reconstruction of Flexible Planer Structures Based on FBG"

_micromachines, 2022, doi:10.3390/mi13081237_

Round 1

Reviewer 1 Report

Dear authors, your work is interesting but I want to clarify some points.

1.       Line 178 – temperature correction for FBG sensors is always required and can cause additional errors. Have you done it somehow?

2.       Line 195. For me such description of WDMT and SDMT looks a little bit strange. FBGs in one line are also separated in space. And I can use 16 different wavelengths to separate 16 FBGs in onу line by WDMT.

3.       Figure 2 – please translate to English. And about scheme – why you haven’t fixed some FBGs in orthogonal Y direction? It can give a lot information to increase accuracy I guess.

4.       Figure 3a – please, change angle of view to show torque arrow.

5.       Line 235. Please, explain why you generate forces randomly instead of number of small initial forces in each coordinate point? Whether ML approach here is more useful then analytical calculation of shape through equations for strain?

6.       Figure 9 – hard to read labels, please make proper background.

7.       Which parameters has your FBG system? Peak detection error? Why 5 outputs are connected to 4 sensing lines?

8.       Figure 11 – please subscribe axis.

Author Response

Point 1:  Line 178 – temperature correction for FBG sensors is always required and can cause additional errors. Have you done it somehow?

Response 1:

During the test, we will set up a separate temperature-compensated light circuit to obtain the central wavelength drift of the FBG due to temperature change. That is, an individual FBG, which is not subject to any forces, is only affected by changes in the temperature of the measurement environment. During strain measurement, we will subtract this wavelength drift due to temperature effects, so we will simplify the temperature change to zero to simplify the derivation of the formula.

Point 2:  Line 195. For me such description of WDMT and SDMT looks a little bit strange. FBGs in one line are also separated in space. And I can use 16 different wavelengths to separate 16 FBGs in onу line by WDMT

Response 2:

We have modified these errors in lines 193-196. See the revised manuscript submitted for details.

Point 3:  Figure 2 – please translate to English. And about scheme – why you haven’t fixed some FBGs in orthogonal Y direction? It can give a lot information to increase accuracy I guess.

Response 3:

Figure 2:We have modified these errors in line 205. See the revised manuscript submitted for details.

In some literature, the measurement points are laid out in the form of orthogonal FBG to obtain the curvature of the measurement points in the two directions of the X and Y axis for curve reconstruction, and this layout method will improve the reconstruction accuracy. We have experimented with this orthogonal layout, which has improved the accuracy of the measurement of the deformation variables of the measurement points, but the effect on the reconstruction accuracy of the flexible structure shape reconstruction is not obvious. We also agree that two FBGs should be the layout for each measurement point, but do not consider the relationship between them to be orthogonal, and are currently looking for ways to confirm the size of the angle between the two FBGs

Point 4: Figure 3a – please, change angle of view to show torque arrow.

Response 4:

We try to rotate the structure because of the characteristics of the division mesh of the Ansys finite element software, when we can see the torque arrow, the structure body will appear jagged pattern, and the structure surface is not smooth. The picture quality will be worse.

Point 5: Line 235. Please, explain why you generate forces randomly instead of number of small initial forces in each coordinate point? Whether ML approach here is more useful then analytical calculation of shape through equations for strain?

Response 5:

Randomly generated forces are applied to the flexible structure to simulate the loading conditions to which the flexible structure is subjected during its use. The above process is to build a complete training data set and collect the strain values and deformation variables of each measurement point under different forces, which is more convenient and faster than a small initial forces in each coordinate point

   Machine learning method, with the ability of active learning, can establish the mapping relationship between the strain value of the measurement point and the shape variable based on a large amount of training data, and can correct the model according to the training data, the formula to deduce the deformation relationship, but does not have the ability of automatic learning, not able to correct the formula according to the test results.

Point 6: Figure 9 – hard to read labels, please make proper background.

Response 6:

We have modified the problem. See the revised manuscript submitted for details.

Point 7:   Which parameters has your FBG system? Peak detection error? Why 5 outputs are connected to 4 sensing lines?

Response 7:

In the FBG system, the parameter of interest is the central wavelength of each FBG. From the drift of the central wavelength of the FBG, we can calculate the value of strain change at each measurement point.

5 outputs are 4 measurement sensing lines and a backup sensing line.

Point 8: Figure 11 – please subscribe axis

Response 8:

We have modified these errors in lines 385. See the revised manuscript submitted for details

Reviewer 2 Report

This paper reports a study about deformation monitoring and shape reconstruction of flexible planer structures based on FBG. I have some comments.

Introduction needs improvements and include some words about FBG for different usages like bio and physical ones. Read and refer: Journal of lightwave technology 31 (10), 1551-1558, 2013; IEEE Sensors Journal 19 (17), 7168-7178, 2019.

In the sentence "The fiber grating 75 sensing measurement method uses diverse forms of fiber grating sensing" - I cannot understand the content. Please improve the english.

Fig. 2 has chinese letters. Please correct.

The authors used many FBG. How can optimize the number of FBGs to reduce the cost without reduce the performance?

How can the signal processing or machine learning algorithm can help to optimize much more the results already achieved with the proposed system and analysis?

Can have other optical technology with similar performance to avoid high costs of FBGs?

Author Response

Point 1: Introduction needs improvements and include some words about FBG for different usages like bio and physical ones. Read and refer: Journal of lightwave technology 31 (10), 1551-1558, 2013; IEEE Sensors Journal 19 (17), 7168-7178, 2019.

Response 1:

These two literatures summarize the FBG application very well. We quoted and reviewed two literatures: “FBG is a popular direction in current sensor research, and its applications are wide [14,15]”

Point 2: In the sentence "The fiber grating 75 sensing measurement method uses diverse forms of fiber grating sensing" - I cannot understand the content. Please improve the English.

Response 2:

In line 75, the sentence was modified to: Fiber grating sensing measurement methods can achieve real-time measurements of flexible planar structures based on diverse fiber grating sensing forms [29-31],

Point 3: Fig. 2 has Chinese letters. Please correct

Response 3:

We have modified the error in line 205. See the revised manuscript submitted for details.  

Point 4: The authors used many FBG. How can optimize the number of FBGs to reduce the cost without reduce the performance?

Response 4:

For the sensor layout problem, we consider particle swarm optimization algorithm, but the sensors layout according to the optimization results are not easy to reconstruct the shape, especially when the curve reconstructs the surface.

Point 5: Fig. 2 has Chinese letters. Please correct How can the signal processing or machine learning algorithm can help to optimize much more the results already achieved with the proposed system and analysis?

Response 5:

We have modified the error in Fig. 2. See the revised manuscript submitted for details.

Machine learning method, with the ability of active learning, can establish the mapping relationship between the strain value of the measurement point and the shape variable based on a large amount of training data, and can correct the model structure to improve the prediction results of the model according to the training data.

Point 6: Can have other optical technology with similar performance to avoid high costs of FBGs?

Response 6:

For flexible structure reconstruction, the camera technique is one of the more common optical technology, which is lower than the cost of FBG. Due to its large amount of measured data, it cannot achieve real-time measurement and shape reconstruction.

Round 2

Reviewer 1 Report

To previous points:

Point 4: Figure 3a – please, make it readable anyway, add a view from another angle or draw arrow in paint. Now it looks useless and unreadable.

Point 7: Which parameters has your FBG system? Peak detection error? Why 5 outputs are connected to 4 sensing lines?

Response 7:

In the FBG system, the parameter of interest is the central wavelength of each FBG. From the drift of the central wavelength of the FBG, we can calculate the value of strain change at each measurement point.

Yes, FBG wavelength is the parameter of interest. Please. include algorithm of its determination and its accuracy.

Author Response

Point 4: Figure 3a – please, make it readable anyway, add a view from another angle or draw arrow in paint. Now it looks useless and unreadable.

Response:

We have modified the problem by drawing arrow in paint and adding the left view.

Point 7: Which parameters has your FBG system? Peak detection error? Why 5 outputs are connected to 4 sensing lines?

Response 7:

In the FBG system, the parameter of interest is the central wavelength of each FBG. From the drift of the central wavelength of the FBG, we can calculate the value of strain change at each measurement point.

Yes, FBG wavelength is the parameter of interest. Please. include algorithm of its determination and its accuracy.

Response:

The interrogator has a built-in tunable laser source, the reflection spectrum is measured by observing optical power while sending different wavelengths into the fiber sequentially. Then a curve fitting method is utilized, which fits the reflection spectrum of a fiber grating using a Gaussian curve. The mathematical expectation (μ) of the Gaussian curve is considered to be the location of the peak of the reflection spectrum.

The fiber Bragg grating interrogator is calibrated using a F-P etalon. The maximum error between the standard etalon wavelength and the interrogated wavelength does not exceed 5pm, therefore, it can be said that the error of the instrument is less than or equal to 5pm.

Reviewer 2 Report

I am happy with the improvements

Author Response

Thank you for your professional review!